# On the Use of Quantitative Sensory Testing to Estimate Central Sensitization in Humans. Comment on Schuttert et al. The Definition, Assessment, and Prevalence of (Human Assumed) Central Sensitisation in Patients with Chronic Low Back Pain: A Systematic Review. *J. Clin. Med.* 2021, *10*, 5931

**DOI:** 10.3390/jcm11071982

**Published:** 2022-04-02

**Authors:** Ole Kudsk Jensen

**Affiliations:** The Spine Centre, University Research Clinic for Innovative Patient Pathways, Silkeborg Regional Hospital, 8600 Silkeborg, Denmark; okj@dadlnet.dk; Tel.: +45-20987452

With great interest, I have read the systematic review in your journal [1] regarding the use of questionnaires and quantitative sensory testing (QST) in non-specific low back pain (NLBP) patients in order to identify the percentage of patients in whom nociplastic pain mechanism may be operating. As central sensitisation is only identifiable with certainty in animal studies, a new concept is launched for this phenomenon in human studies, i.e., Human Assumed Central Sensitisation (HACS). According to the review, the prevalence of HACS is 43.2% in patients with NLBP, albeit with wide variation (13–78%).

Digital tender point (TP) testing was one of the first QSTs introduced to clinical practice [2]. It is strange, then, that this type of testing was not included in the review. Originally, TP examination was used in research primarily to distinguish fibromyalgia patients from patients with inflammatory rheumatologic disorders. The test was performed by applying a standardized pressure with the thumb that is gradually increased by 1 kg/s. up to 4 kg. The technique was trained using a dolorimeter [3]. Eighteen locations, symmetrically distributed across the body, were selected after a careful process. The single point was counted as positive if the pressure resulted in pain. In 1990, after analyzing data from 558 rheumatologic patients, fibromyalgia was defined as widespread pain for more than 3 months and more than 10 of 18 TPs sore on pressure [3].

Thus, the result of TP examination is the number of painful points induced by a pressure of ≤4 kg. Although the result of testing every single point is dichotomous, the TP count reflects the degree of diffuse pressure tenderness in the body; i.e., it is a measure of global mechanical hyperalgesia in the range of 0–18.

The validity of TP examination has been questioned [2], and fibromyalgia is nowadays usually diagnosed using a questionnaire supplemented with clinical assessment [4]. However, fibromyalgia can still be diagnosed using the original criteria defined in 1990 [4].

TP examination has been used as part of QST in population studies and fibromyalgia studies [5,6,7,8,9,10,11]. In general, TPs are associated with pain intensity, psychological distress, and disability. It has also been shown that widespread pain patients with >10 TPs have more pain and disability than widespread pain patients with 0–10 TPs [12].

Apart from our studies [13,14,15], TP examination has only been used as part of QST in a few NLBP studies [16,17,18]. However, we have studied TP associations in LBP patients more rigorously. We have shown that the TP count is positively associated with back pain intensity and bodily distress and negatively associated with disc degeneration on X-rays, and it is also negatively associated with radiculopathy [13]. We have also studied the reproducibility and the reliability of TP examination in chronic LBP patients, and we found both estimates to be acceptable; agreement was +/−3 TPs in more than 70% of tests, and reliability was 0.72–0.84 [14].

Finally, we have shown that the TP count in low back patients is negatively associated with most types of degenerative changes shown via magnetic resonance imaging (MRI) of the lumbar spine. However, men with >7 TPs and women with >10 TPs had higher back pain intensity than patients with few TPs, in spite of having fewer degenerative changes shown via MRI compared to patients with few TPs [15]. Accordingly, men with >7 TPs and women with >10 TPs had disproportionate back pain, since back pain was not explained by degenerative changes. These patients comprised 44% of the NLBP patients, which is, therefore, our estimate of HACS in NLBP.

## Data Availability

Not applicable.

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
