# Peer review of "On the Use of Quantitative Sensory Testing to Estimate Central Sensitization in Humans. Comment on Schuttert et al. The Definition, Assessment, and Prevalence of (Human Assumed) Central Sensitisation in Patients with Chronic Low Back Pain: A Systematic Review. J. Clin. Med. 2021, 10, 5931"

_jcm, 2022, doi:10.3390/jcm11071982_

Round 1

Reviewer 1 Report

The author presents a comment on the recently published systematic review "The definition, assesment, and prevalence of (human assumed)" Central sensitization in patients with chronic low back pain: a systematic review".

He basically wonders why the authors of the systematic review have not included Tender Points examination in the review process.

He states "As digital tender point (TP) examination also may be recognized as a QST method". Can the author provide references in which Tender Point examination is considered part of a QST protocol? Digital TP examination does not use any cuantitative variable: just the number of tender points, which is a sum of several dichotomical variables. How therefore it coul be considered part of the quantitative sensory testing?

The author should provide further reasons and elaborate on the idea of considering digital tender point examination a part of the QST.

Author Response

Reviewer’s comment:

The author presents a comment on the recently published systematic review "The definition, assesment, and prevalence of (human assumed)" Central sensitization in patients with chronic low back pain: a systematic review".

He basically wonders why the authors of the systematic review have not included Tender Points examination in the review process.

He states "As digital tender point (TP) examination also may be recognized as a QST method". Can the author provide references in which Tender Point examination is considered part of a QST protocol? Digital TP examination does not use any cuantitative variable: just the number of tender points, which is a sum of several dichotomical variables. How therefore it coul be considered part of the quantitative sensory testing?

The author should provide further reasons and elaborate on the idea of considering digital tender point examination a part of the QST.

Response:

I would like to thank the reviewer for comments which hopefully have improved my letter, please see the attachment.

Answer: In the second section, I have quoted one of the leading pain researchers in the world with the phrase: “Digital tender point testing was one of the first QSTs introduced to clinical practice (Staud et al. 2012)” Afterwards, I have briefly explained the historical background for developing the use of tender point examination in diagnosing fibromyalgia. In addition, I have tried to explain how 18 dichotomous variables can result in a global measure for mechanical hyperalgesia. Furthermore, I have listed references to studies using tender point examination as a quantitative sensory testing method. Finally, I have made minor changes in the text referring to our own research in low back pain patients.

Sincerely                                                                                                                     Ole Kudsk Jensen

Round 2

Reviewer 1 Report

The author has implemented changes according to the commentaries.